# Calorimetric evidence for two phase transitions in $Ba_{1−x}K_xFe_2As_2$ with fermion pairing and quadrupling states

Ilya Shipulin[1,2,11], Nadia Stegani[3,4,11], Ilaria Maccari[5,6], Kunihiro Kihou[7], Chul-Ho Lee[7], Quanxin Hu[8], Yu Zheng[8], Fazhi Yang[8], Yongwei Li[8], Chi-Ming Yim[8,9], Ruben Hühne[1], Hans-Henning Klauss[2], Marina Putti[3,4], Federico Caglieris[3,4,10] ✉, Egor Babaev[5] ✉ & Vadim Grinenko[8,9] ✉

Materials that break multiple symmetries allow the formation of four-fermion condensates above the superconducting critical temperature ($T_c$). Such states can be stabilized by phase fluctuations. Recently, a fermionic quadrupling condensate that breaks the $Z_2$ time-reversal symmetry was reported in $Ba_{1−x}K_xFe_2As_2$. A phase transition to the new state of matter should be accompanied by a specific heat anomaly at the critical temperature where $Z_2$ time-reversal symmetry is broken ($T_c^{Z2} > T_c$). Here, we report on detecting two anomalies in the specific heat of $Ba_{1−x}K_xFe_2As_2$ at zero magnetic field. The anomaly at the higher temperature is accompanied by the appearance of a spontaneous Nernst effect, indicating the breakdown of $Z_2$ symmetry. The second anomaly at the lower temperature coincides with the transition to a zero-resistance state, indicating the onset of superconductivity. Our data provide the first example of the appearance of a specific heat anomaly above the superconducting phase transition associated with the broken time-reversal symmetry due to the formation of the novel fermion order.

The formation of bound states of fermions can lead to new states of matter: symmetry-breaking condensates. The most investigated case is the condensate of paired electrons (Cooper pairs). It results in a new state of matter: superconductivity, characterized by a spontaneously broken $U(1)$ gauge symmetry[1–3]. A complex order parameter $\Delta$ describing the simplest single-band spin-singlet superconducting state is associated with non-vanishing averages of two-fermion annihilation operators $<c_\uparrow c_\downarrow>$. It describes the flow of Cooper pairs, each carrying twice the electron charge "$e$". Bound states of 4, 6 etc. electrons would also be bosons, but within the standard Bardeen-Cooper-Schrieffer (BCS) theory, such condensates do not form. Four-fermion electronic condensates can appear via a fluctuations-driven mechanism if the system breaks multiple symmetries[4–9]. Among four-fermion orders, there is a counterpart of Cooper pair superconductivity: a charge-4e superconducting order parameter can be constructed involving nonzero averages of the kind $<c_{\uparrow i}c_{\downarrow i}c_{\uparrow j}c_{\downarrow j}>$, where $i$ and $j$ are component indices. A bosonic counterpart of such states was also discussed in the context of an ultracold atomic mixture close to the Mott insulating state[10], and in a mixture of neutral and charged fermions[4].

[1]Institute for Metallic Materials, Leibniz-IFW Dresden, D-01069 Dresden, Germany. [2]Institute for Solid State and Materials Physics, Technische Universität Dresden, D-01069 Dresden, Germany. [3]University of Genoa, Via Dodecaneso 33, 16146 Genoa, Italy. [4]Consiglio Nazionale delle Ricerche (CNR)-SPIN, Corso Perrone 24, 16152 Genova, Italy. [5]Department of Physics, KTH Royal Institute of Technology, SE-106 91 Stockholm, Sweden. [6]Laboratory for Theoretical and Computational Physics, Paul Scherrer Institute, 5232 Villigen PSI, Switzerland. [7]National Institute of Advanced Industrial Science and Technology (AIST), Tsukuba, Ibaraki 305-8568, Japan. [8]Tsung-Dao Lee Institute, Shanghai Jiao Tong University, Shanghai 201210, China. [9]School of Physics and Astronomy, Shanghai Jiao Tong University, Shanghai 200240, China. [10]Institute for Solid State Research, Leibniz-IFW Dresden, D-01069 Dresden, Germany. [11]These authors contributed equally: Ilya Shipulin, Nadia Stegani. ✉e-mail: federico.caglieris@spin.cnr.it; babaev@kth.se; vadim.grinenko@sjtu.edu.cn

This type of ordering opens an interesting research avenue especially because, besides the superconducting states of higher composites, one can construct a great diversity of orders out of four-fermionic operators. The variety of the potential new states described by such order parameters is much greater than possible orders arising from fermionic pairs. An especially interesting possibility is associated with fermion quadrupling condensates forming above the superconducting phase transition, leading to condensates with principally different properties than superconductors. One such possible state appears when a fermion quadrupling condensate results in a Broken Time-Reversal Symmetry (BTRS) [6,11–15]. At low temperatures, such a system is a multicomponent superconductor that breaks time-reversal symmetry and can be described by several complex fields $\Delta_i$, where $i$ is a component or band index (see e.g., [16–20]). Since applying the time-reversal operation twice returns the system to its original state, such a condensate breaks an additional twofold (i.e., $Z_2$) symmetry. A new state of matter forms under temperature increase [12]. In this state, there is no order in the pairing fields in each band, $\langle \Delta_i \rangle = 0$, but there appears a non-vanishing composite order parameter $\Delta_{4f} \propto \langle \Delta_i^* \Delta_j \rangle \neq 0$ [6,11,12,14]. This order parameter is of fourth order in fermionic fields $<c_{\uparrow i} c_{\downarrow i} c_{\uparrow j}^\dagger c_{\downarrow j}^\dagger>$. The state preserves the local $U(1)$ symmetry and hence it is resistive to dc current. Instead, it breaks $Z_2$ time-reversal symmetry resulting in dissipationless local counterflows of charges between $i$ and $j$ components. These currents produce spontaneous magnetic fields around certain kinds of inhomogeneities and topological defects [12,13]. Many novel properties of this state can be described by an effective model, which is different from the Ginzburg-Landau effective models of superconductors and the Gross-Pitaevskii effective models of superfluids, but is rather related to the Skyrme model [12,13]. The predicted weak spontaneous magnetic fields appearing at $T_c^{Z2}$, above the superconducting critical temperature $T_c$, were detected in recent $\mu$SR and spontaneous Nernst effect measurements in the $Ba_{1-x}K_xFe_2As_2$ system [12,21]. This, and other experimental data in ref. 12 provides the evidence for a four-fermionic (quartic) state, which exists in a range of temperatures above $T_c$. The recent scanning SQUID study identified fractional flux vortices that were theoretically expected in the systems with fermion quadrupling phase [22].

A transition from a normal state to a new state, such as the quartic state, should result in a specific heat anomaly at $T_c^{Z2}$. The anomaly should be small since it is associated with the order in the phase difference of the components. A second anomaly at $T_c$, associated with the order in the sum of the component's phases, should follow at a lower temperature. An example of these anomalies from Monte-Carlo simulations of a multiband model is shown in the theoretical analysis section below. However, the predicted two anomalies are expected to be difficult to detect experimentally since the quartic phase is a fluctuation-induced effect, in which phase fluctuations in the system that we consider, cause a tiny contribution on top of a background due to other degrees of freedom. The phase-fluctuations contribution, in addition, can be washed out by inhomogeneities since $T_c^{Z2}$ strongly depends on doping [21]. Such double anomalies in zero external fields have not observed in experiments so far. Here, we investigate new samples of $Ba_{1-x}K_xFe_2As_2$ with slightly different doping compared to the one studied in ref. 12 and report the observation of two anomalies in the zero-field specific heat. At this doping, the low-temperature superconducting state breaks time-reversal symmetry. Theoretical modeling presented evidence that this low-temperature state has $s+is$ symmetry [16–19,21,23], or similar states such as $s+id$ [20]. Importantly, these anomalies are consistent with spontaneous Nernst and electrical resistivity data, signaling two separate $Z_2$ and $U(1)$ phase transitions. These data provide calorimetric evidence for a phase transition above $T_c$ associated with the formation of the quartic state.

## Results

### Experimental results

The temperature dependence of the specific heat and the magnetic susceptibility measured on the sample ($S_{NP}$) from ref. 12 is shown in Fig. 1e. There is a significant splitting between the onset temperatures for the specific heat anomaly and the diamagnetic susceptibility. As discussed in ref. 12, this splitting is related to the precursor formation of electronic bound states and eventually to the $Z_2$ phase transition above $T_c$. In ref. 12 and in the current study, the transition temperature $T_c^{Z2}$ is defined by the onset of the spontaneous Nernst effect signal that requires a spontaneous breaking of time-reversal symmetry. In turn, superconducting $T_c$ is determined by the temperature at which electrical resistivity is zero. It is seen in Fig. 1e that $T_c$ defined in this way corresponds to the maximum in the specific heat. However, the expected distinct anomaly in the specific heat at $T_c^{Z2}$ was not resolved in zero external field in samples investigated in ref. 12. In this work, we performed systematic specific heat measurements of several new samples ($S2–S4$) with a doping level of $x \approx 0.75$, close to $x = 0.77$ for the $S_{NP}$ sample. According to the previous studies [21,24], this doping level is within the range where the superconducting state breaks time-reversal symmetry. In addition, we had two reference samples ($S1$, and $S5$) that do not break time-reversal symmetry. The specific heat data $\Delta C_{el}/T$ are summarized in Fig. 1. The raw data are given in Supplementary Information Fig. S1 for samples $S1$, $S3–S5$, and in Fig. 2b for sample $S2$.

The reference samples (Fig. 1a, f) show conventional behavior with a single-phase transition at $T_c$ expected for standard superconductors: the appearance of non-dissipative currents observed in the susceptibility data match with the onset of the specific heat anomaly. The samples with the BTRS superconducting state (b–d)) show qualitatively different behavior. Similarly to the previously investigated $S_{NP}$ sample shown on e, these samples exhibit the fermion quadrupling phase. However, in contrast to sample $S_{NP}$, the samples investigated in this work show a step-like anomaly above $T_c$. We compare the susceptibility and specific heat in Fig. 1 to illustrate the different behavior between samples with and without BTRS state. In a small part of a sample, a short-range superconducting order may appear at a slightly higher temperature than bulk; hence, in a fluctuating/slightly inhomogeneous system, the onset of the diamagnetic signal in susceptibility could appear already above bulk $T_c$. Some diamagnetic signal in the susceptibility at temperatures above zero resistivity is observed for $S_{NP}$ (Fig. 1e) and S2 samples (Fig. 1b). However, the additional anomalies in the specific heat cannot be caused by superconducting phases surviving to a higher temperature since no superconducting signal is observed in the magnetic susceptibility at the corresponding temperature (right axis in Fig. 1). For a detailed analysis of a possible inhomogeneity effect on the susceptibility and specific heat, see ref. 12.

To investigate whether this anomaly can be associated with the $Z_2$ time-reversal-symmetry-breaking phase transition, we performed more detailed investigations on sample S2, which has the most pronounced double anomalies in the specific heat. It is important to emphasize that the size of the anomaly related to the ordering of the phase difference is small for all samples compared to the broad background due to the formation of incoherent Cooper pairs (for detailed discussion, see section Theoretical analysis). The data for S2 sample are summarized in Fig. 2 and Fig. S2. Figure 2a shows the electrical resistivity plotted versus squared temperature. The resistivity follows a conventional Fermi liquid behavior in the normal state with the residual resistivity ratio $RRR \approx 70$. This indicates high sample quality and no evidence of proximity to critical points of magnetic or charge density wave (CDW) orders. To further characterize the state, we examined the surface of the $S_{NP}$ sample using scanning transmission microscopy (STM). The data are shown in Supplementary Information, Fig. S3. Our measurements gave no indication of CDW or spin density wave (SDW) at low temperatures, neither in the topography

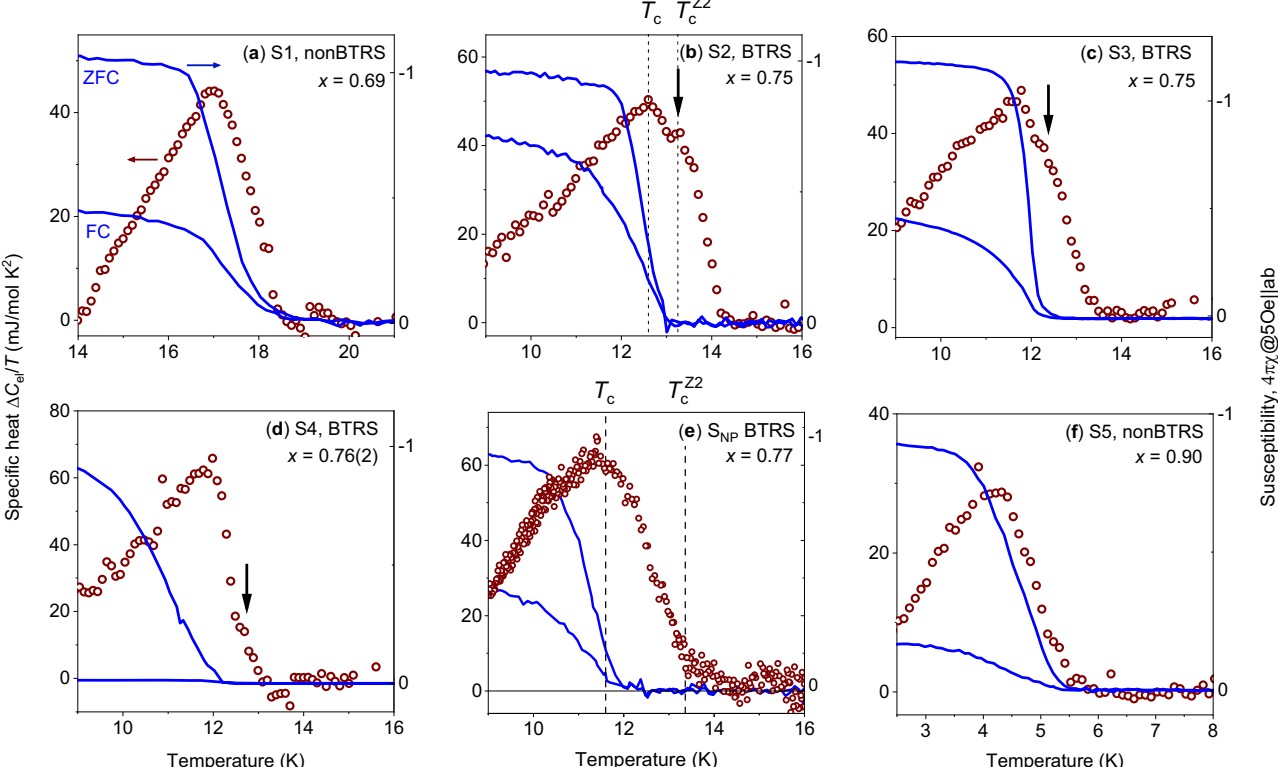

**Fig. 1 | Split transitions in hole overdoped $Ba_{1-x}K_xFe_2As_2$.** Temperature dependence of the zero-field specific heat $\Delta C_{el}/T$ and the static magnetic susceptibility measured in $B\|ab = 0.5$ mT. **e** shows the data for the sample from ref. [12] with the quartic metal phase and doping level $x = 0.77$. $T_c$ is defined by zero resistance and $T_c^{Z2}$ is shown according to the onset of the spontaneous magnetic fields and spontaneous Nernst effect. A distinct specific heat anomaly at $T_c^{Z2}$ was not resolved within error bars of the measurements. **a**–**d**, **f** show the data for the samples studied in this work. **a**, **f** correspond to reference samples with doping levels $x \approx 0.69$ and

$x \approx 0.90$, respectively. These two samples have a doping level outside the range of doping, where BTRS occurs[21,24] show the conventional picture, where the onset of the specific heat is consistent with the onset of a superconducting response in the susceptibility. This behavior is indistinguishable from the standard mean-field behavior of a superconductor with a single-phase transition at $T_c$. **b**–**d** are for the samples with BTRS state and doping level $x \approx 0.75$. These samples show two anomalies in the specific heat. The anomaly at a higher temperature, highlighted by black arrows, appears at the $Z_2$ transition (see Fig. 2).

nor in $dI/dV − V$ spectra. The inset in **a** shows the difference between the experimental data and the $T^2$ fitting curve. It is seen that resistivity deviates from $T^2$ behavior below the characteristic temperature $T_{SCF} \approx 18$ K. We define $T_{SCF}$ as the crossover associated with the onset of detectable effects of superconducting fluctuations[12]. Note that, compared to ~10 K for a sample $S_{NP}$, here the temperature difference between $T_{SCF}$ and $T_c \approx 12.6$ K is smaller. This is consistent with the scenario where this state arises on top of the $s + is$, $s + id$, or similar pairing that in usual models exists for a very narrow range of doping.

The specific heat data measured in zero and applied magnetic field $B\|c = 16$ T are shown in Fig. 2b. The applied magnetic field did not completely suppress the specific heat anomaly associated with the formation of incoherent Cooper pairs, as expected from the field-temperature phase diagram obtained in ref. [12]. Therefore, to subtract the phonon contribution from the zero-field specific heat, we fitted the in-field specific heat above 10 K and used the fitting curve as the phonon background for zero-field data. Details of the fitting procedure can be found in refs. [21,24]. The result is shown in **c**, left axis, and compared with the temperature dependence of the zero-field spontaneous Nernst effect, right axis. The onset of the spontaneous Nernst signal gives the critical temperature of the BTRS transition at $T_c^{Z2} \approx 13.25$ K, indicating the formation of the four-fermion-ordered state characterized by intercomponent phase-difference locking. This transition takes place at a significantly lower temperature than the crossover associated with superconducting fluctuations $T_{SCF}$. That is consistent with the theoretical expectations that the relative phase

ordering of the four-fermion order parameter $\langle \Delta_i^* \Delta_j \rangle \neq 0$ should occur below the onset of incoherent pairing fluctuations. The Nernst signal vanishes below $T_c$ when fluctuations are suppressed and the transverse voltage is zero. Thus, it is natural to expect in these samples a maximum in the Nernst signal somewhere more or less close to $T_c$ due to the increase of the $Z_2$ order parameter with the decreasing temperature. The shape of the temperature dependence of the Nernst signal should not be universal since it depends on the size of the splitting between $T_c$ and $T_c^{Z2}$. The most relevant characteristic is the onset of the spontaneous Nernst signal[12]. That effect requires a spontaneous breaking of time-reversal symmetry. Remarkably, as shown in **c**, the $T_c^{Z2}$ transition temperature coincides with the high-temperature anomaly in the specific heat.

Next, our data shows that $T_c^{Z2}$ splits from the superconducting critical temperature $T_c$. For this purpose, the temperature dependence of the zero-field electrical resistivity (left axis) is compared with the Seebeck effect (right axis) in **d** of Fig. 2. Both quantities are signaling superconducting phase transitions at $T_c \approx 12.6$ K defined by the temperature at which the resistivity and the Seebeck effect are zero. This temperature is lower than $T_c^{Z2} \approx 13.25$ K. Notably, $T_c$ coincides with the lower temperature anomaly (sharp maximum) in the specific heat, indicating that the appearance of zero resistance is caused by the appearance of a superconducting order associated with electron pairs[12]. Note that the double-step anomaly in the zero-field specific heat is well visible in the raw data shown in **b**. These observations allow us to conclude that our data provide calorimetric evidence for a $Z_2$ phase transition above $T_c$.

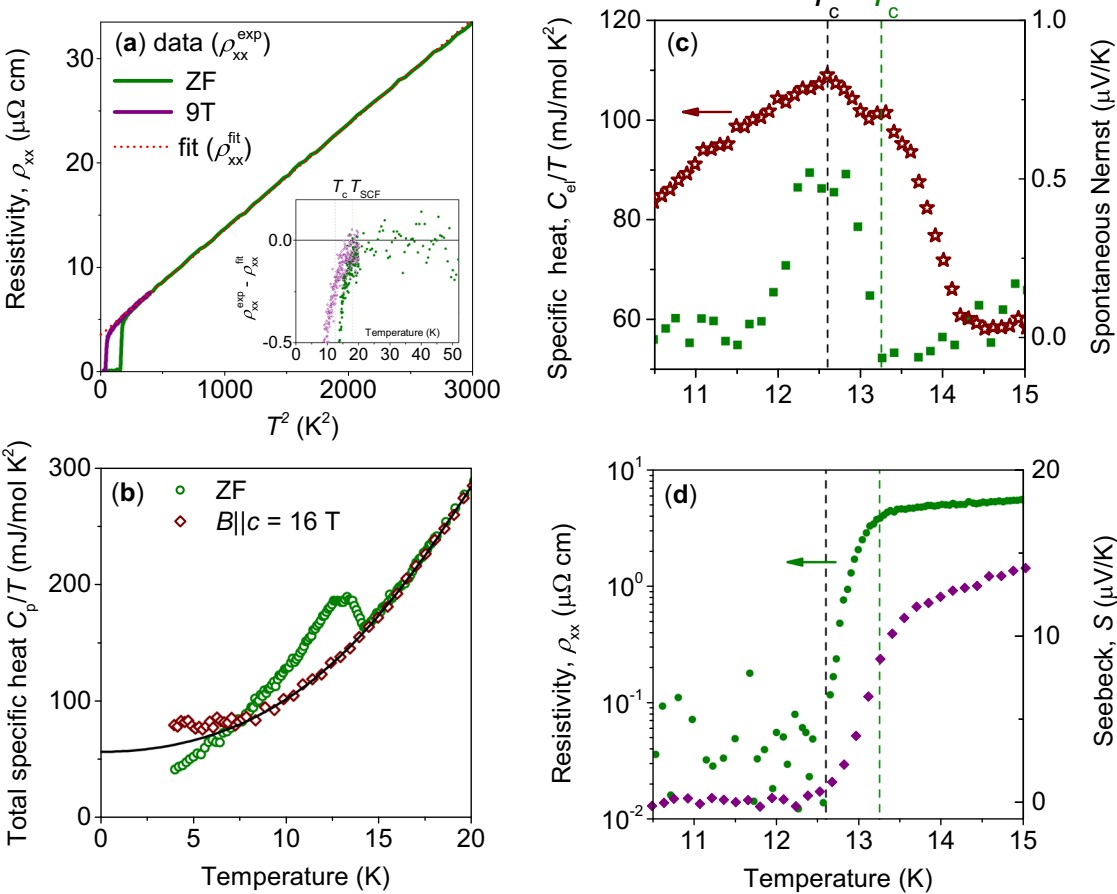

**Fig. 2 | Two phase transitions in Ba$_{1-x}$K$_x$Fe$_2$As$_2$.** The figure shows the experimental data for S2 sample. **a** The longitudinal electrical resistivity vs. temperature squared measured in zero and $B\|c$ = 9 T field applied along the crystallographic $c$-axis. The inset shows the temperature dependence of the difference between the fit curves and the experimental data. The resistivity deviates from the normal state behavior at the crossover temperature $T_{SCF}$ associated with the onset of superconducting fluctuations. **b** Temperature dependence of the specific heat measured in zero and $B\|c$ = 16 T field. The solid line is a fit to approximate the normal state behavior. A two-step anomaly is seen in the zero-field specific heat data. **c** (left axis) Temperature dependence of the difference between the zero-field specific heat experimental data and the phonon contribution defined using the 16 T experimental data from **b** compared with (right axis) temperature dependence of the zero-field spontaneous Nernst effect. The appearance of the spontaneous Nernst effect signals a spontaneous breaking of time-reversal symmetry at $T_c^{Z2}$. It coincides with the high-temperature anomaly in the specific heat. **d** Temperature dependence of the zero-field electrical resistivity (left axis) and the Seebeck effect. The specific heat in **c** shows the second anomaly appearing at a lower temperature corresponding to the superconducting $T_c$.

## Theoretical analysis

To demonstrate the two singularities in the specific heat, associated with ordering of the phase difference and phase sums at different temperatures, we use the simplest phenomenological two-component free-energy functional that yields a similar phase diagram (for detail, see ref. 13,14,25). We use a simple London model with two phases $\phi_{1,2}$ originating from two complex fields $\Psi_\alpha = |\Psi_\alpha|e^{i\phi_\alpha}$. The microscopic relation between these fields and parameters of $s + is$ or $s + id$ superconductors can be found in ref. 25. These fields are proportional to the order parameter values $\Delta_i$ at different bands or their linear combinations. The existence of a fermion quadrupling phase in that model was previously reported in ref. 12,14 without discussing the specific heat.

The model

$$f = \frac{1}{2}\sum_{\alpha=1,2}\left(\nabla\phi_\alpha\right)^2 - \nu(\nabla\phi_1 \cdot \nabla\phi_2) + \eta_2\cos[2(\phi_1 - \phi_2)]. \quad (1)$$

accounts for two different inter-component interactions: the second-order biquadratic Josephson interaction with a coupling constant $\eta_2 > 0$, and the mixed-gradient term with a coupling constant $\nu > 0$. The presence of the biquadratic term reduces the total symmetry of the model to $U(1) \times Z_2$, where the $Z_2$ time-reversal symmetry is associated

with the two equivalent minima for the phase-difference mode, i.e. $\phi_{1,2} = \phi_1 - \phi_2 = \pm\pi/2$, spontaneously broken at low temperatures. This represents one of the simplest models with a BTRS fermion quadrupling phase in zero external magnetic field[14]. Here, we consider the model situation where we neglect the coupling to an electromagnetic field, i.e. assuming infinite magnetic field penetration length. Finite values of the magnetic field penetration length increase the size of the fermion quadrupling phase[14].

In this work, we discuss the specific heat of the system in the presence of a finite splitting between the two critical temperatures of the order of 10%, similar to the splitting experimentally observed. In multicomponent models, when the splitting between the transitions is small, a single transition could become weakly first order[26,27], which would alter the specific heat picture. In our case, the splitting is relatively large, and as shown in Fig. 3, our Monte-Carlo (MC) simulations reveal that the specific heat $C_v$ develops two anomalies in the proximity of the two critical temperatures $T_c$ and $T_c^{Z2}$, respectively. We note that the model we consider is a phase-only model that only accounts for the contribution of the phase fluctuations to the specific heat. In a real system, such as Ba$_{1-x}$K$_x$Fe$_2$As$_2$, this will represent small additional contributions superimposed with the non-singular smooth contributions from pair-breaking effects.

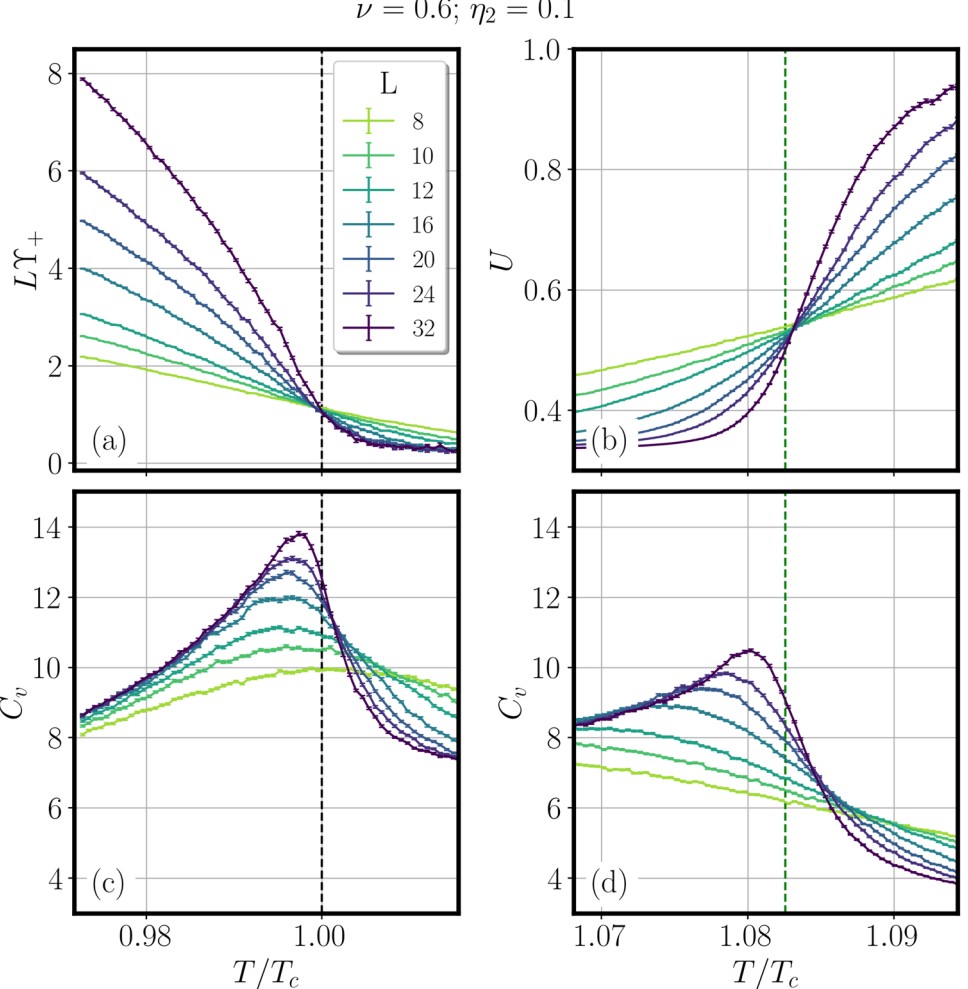

**Fig. 3 | An example of double specific heat anomalies. a, b** The results from Monte-Carlo computations in the simplest London model with electron quadrupling BTRS state from refs. 12,14 for different system sizes $L$ (model description can be found in the Methods section). **a** Helicity-modulus $\Upsilon_+$ as a function of the temperature, signaling the superconducting phase transition at $T_c$. **b** Binder cumulant $U$ versus $T/T_c$ signaling the breakdown of the time-reversal symmetry at $T_c^{Z2}$. **c, d** The calculated specific heat as function of $T/T_c$, for temperatures close to the $T_c$ (**c**) and $T_c^{Z2}$ (**d**) transition temperatures in zero magnetic field. The black and the green dashed lines indicate respectively the $T_c$ and the $T_c^{Z2}$ calculated in the thermodynamic limit as explained in the Methods section. The model gives the contribution to the specific heat from the phase fluctuations only. In a real system such as $Ba_{1-x}K_xFe_2As_2$, this contribution should be superimposed with the non-singular contribution from pair-breaking effects.

Note that Monte–Carlo calculations are done on finite-size systems, and real superconductors always have some degree of inhomogeneity. Hence, for both transitions, instead of a specific heat divergence, one observes anomalies in the form of finite "bumps". The aim of these calculations is to assess the relative size of two bumps coming from the ordering of relative phase sums and relative phase differences, on top of the non-singular contribution associated with pair formation. The obtained ratio between amplitudes of the anomalies is relatively similar to the experimental observations for similar splitting between $T_c$ and $T_c^{Z2}$ transition temperatures.

## Discussion

A specific heat singularity occurs when a system transitions into another state. The electron quadrupling phases of the kind that we discuss in this paper are stabilized by fluctuations. The experimental observation of a specific heat anomaly associated with phase fluctuations is usually difficult in superconducting systems due to the relatively small value of the contribution and also because of the smearing of the transitions by inhomogeneities. For example, so far, the critical exponents of the 3D XY model could not be resolved in superconductors. However, deviations from the mean-field jump and the existence of anomalies associated with phase-fluctuation-driven transitions, i.e., vortex lattice melting, were observed[28–30]. Similar to our experiments, the size of the vortex lattice melting anomaly is very sensitive to the sample quality. The size of the anomaly differs significantly even for the samples with nearly the same $T_c$[28,29]. Resolving phase-fluctuation-driven anomaly is an especially challenging experiment in our case since the splitting between the $Z_2$ and $U(1)$ phase transitions and the strength of superconducting fluctuations are very sensitive to small doping variation, as observed in this study and consistent with the discussion in[11,12]. Time-reversal-symmetry-breaking transition has an extremely sharp dependence on doping even in mean-field theories[16,17]. Here, we found that 2% changes in doping level narrow the splitting by about three times. The strongest splitting of ~1.8 K was observed for $S_{NP}$ sample from ref. 12 with $x = 0.77$, and it reduced to 0.7 K for $x = 0.75$. Therefore, the visibility of the $Z_2$ anomaly for a particular sample is defined by the details of $T_c^{Z2}$ dependence on doping and the location of the anomaly relative to the pair-breaking background.

In a broader context of the calorimetry of a superconducting state, observation of two specific heat anomalies is exceptionally rare. Two specific heat anomalies do not necessarily require an additional

fermion quadrupling phase but should also occur in cases where there are transitions from one electronic phase to another one. Double-anomaly in specific heat is also expected in the more conventional cases in which a system is an ordinary superconductor at a higher temperature and breaks time-reversal symmetry at a lower temperature. Evidence of such superconducting systems exists from multiple measurements (for a recent review, see ref. 31). But almost all attempts to resolve double-anomaly in specific heat were not successful. The most known example of the challenges in resolving thermodynamic signatures of $Z_2$ transition inside the superconducting phase is given by $Sr_2RuO_4$. Recent $\mu$SR measurements under uniaxial stress revealed that superconducting and $T_c^{Z2}$ transition temperatures split under stress[32]. The absence of noticeable superconducting fluctuations in $Sr_2RuO_4$ suggests the conventional mean-field transition between two superconducting states, i.e., $T_c^{Z2} \leq T_c$ as observed in the experiment. In this case, a second anomaly in the specific heat was expected at $T_c^{Z2}$. It was noted that an anomaly may be small in certain models[33,34]. The measurements of the specific heat under uniaxial stress[35], and complementary measurements of the electrocaloric effect[36] did not reveal any noticeable second anomaly at the expected BTRS transition. So far, to the best of our knowledge, two specific heat anomalies associated with superconducting and $Z_2$ transitions were clearly observed only in $UPt_3$, where it was observed below superconducting phase transition[37].

Similar fermion quadrupling condensates may, in principle, form in the presence of pair density wave (PDW) physics see ref. 38. But the analysis of spontaneous fields and the fact that our state exists in a very narrow doping range makes this possibility unlikely. Another evidence in favor of the $s+is$ (or $s+id$, or similar) scenario is provided by the observation of vortices that carry a strongly temperature-dependent fraction of flux quantum[22]. Nonetheless, we examined the possibility of the PDW phase directly. To that end, we performed measurements of the $S_{NP}$ sample surface using STM and found no evidence for the PDW phase (see method section STM and Supplementary Information, Fig. S3).

The main result of this study is that in high-quality $Ba_{1-x}K_xFe_2As_2$ samples, we observed two specific heat anomalies. One correlates with the onset of superconductivity, while the other coincides with the spontaneous breaking of time-reversal symmetry, detected by the appearance of a spontaneous Nernst effect. The breaking of the $Z_2$ symmetry above the superconducting transition temperature and its dependence on doping and external magnetic field allowed earlier to establish that it is associated with the formation of a fermionic quadrupling order[12]. The current data provide calorimetric evidence for the existence of this novel phase in the $Ba_{1-x}K_xFe_2As_2$ system in a zero magnetic field. The second result of this study is the verification of the fermion quadrupling order at a different doping from the one reported in ref. 12.

## Methods
### Samples
Phase purity and crystalline quality of the plate-like $Ba_{1-x}K_xFe_2As_2$ single crystals were examined by X-ray diffraction (XRD) and transmission electron microscopy (TEM)[12,21,24]. The K doping level $x$ of the single crystals was determined using the relation between the $c$ axis lattice parameter and the K doping obtained in previous studies[39]. The selected single-phase samples had a mass ~0.1–1 mg with a thickness ~10–50 μm and a surface area of several mm$^2$.

### Experimental
DC susceptibility measurements were performed using a commercial superconducting quantum interference device (SQUID) magnetometer from Quantum Design. The measurements of the specific heat using the thermal relaxation method were performed in a Quantum Design physical property measurement system (PPMS). The Nernst and Seebeck effect measurements were performed using a home-made sample holder for transport properties inserted in a Quantum Design PPMS endowed with a 9 T magnet. The Seebeck coefficient ($S_{xx}$) is the ratio of the longitudinal electric field to the longitudinal thermal gradient applied to generate it. The Nernst coefficient ($S_{xy}$) is related to the transverse electric field produced by a longitudinal thermal gradient[40]. In order to create an in-plane thermal gradient on the bar-shaped samples, a resistive heater ($R = 2.7$ kΩ) was connected on one side of the sample, while the other side was attached to a thermal mass. The temperature gradient was measured using a Chromel-Au-Chromel differential thermocouple, calibrated in a magnetic field, attached to the sample with a thermal epoxy (Wakefield-Vette Delta Bond 152-KA). The Nernst and Seebeck signals were collected using two couples of electrodes (made of silver wires bonded to the sample with silver paint), aligned perpendicular to or along the thermal gradient direction, respectively. The magnetic field $B$ was applied in the out-of-plane direction along the crystallographic $c$ axis. In order to separate the standard Nernst effect $S_{xy}$ from the spurious Seebeck component (caused by the eventual misalignment of the transverse contacts), the Nernst signal has been antisymmetrized by inverting the $B$ direction. The spontaneous Nernst signal, which is finite only in proximity to the superconducting transition, has been obtained by subtracting the Seebeck ($S_{xx}$) component from the $B$-symmetric part of the Nernst signal as described in ref. 12.

In the thermoelectric measurements, the temperature difference $\Delta T_{sample}$ across the sample (measured by the thermocouple) did not exceed 3% of the measurement temperature $T$ fixed by the thermal mass.

### STM/S measurements
The STM/S experiments were performed using a commercial *Unisoku* USM1300 low-temperature STM machine that operates at a base temperature of 300 mK. Pt/Ir tips were used, and conditioned by field emission with a gold target. Differential conductance (d$I$/d$V$) spectra were recorded using a standard lock-in technique, with the frequency of the bias modulation set at 973 Hz. To obtain clean surface for STM measurements, $Ba_{1-x}K_xFe_2As_2$ with $x = 0.77$ samples were cleaved in situ at ~78 K in ultrahigh vacuum (base pressure $\approx 2 \times 10^{-10}$ mbar). The reported results shown in Fig. S3 were obtained at a sample temperature of 300 mK. The measurements do not show CDW or PDW orders. The red curve in Fig. S3b was a fitting result using the double-gap Dynes equation as described in ref. 41.

$$\frac{dI}{dV} \propto A \int_0^{2\pi} \int_{-\infty}^{\infty} \mathrm{Re}\left[ \frac{|E - i\Gamma_1|}{\sqrt{(E - i\Gamma_1)^2 - (\Delta_1(\cos(4\theta)+1))^2}} \right]$$
$$\times \left( \frac{\exp[(E + eV)/kT]}{kT\{1 + \exp[(E + eV)/kT]\}^2} \right) dEd\theta +$$
$$(1 - A) \int_0^{2\pi} \int_{-\infty}^{\infty} \mathrm{Re}\left[ \frac{|E - i\Gamma_2|}{\sqrt{(E - i\Gamma_2)^2 - \Delta_2^2}} \right] \times \left( \frac{\exp[(E + eV)/kT]}{kT\{1 + \exp[(E + eV)/kT]\}^2} \right) dEd\theta$$
$$(2)$$

In the equation, $\Delta_1$ ($\Delta_2$) and $\Gamma_1$ ($\Gamma_2$) represent the size and broadening terms of the inner (outer) gap. $A$ represents the weight of the nodal superconducting gap. Here, we adopted a nodal gap that was advocated by the ARPES and the penetration depth measurements for this doping range[42,43]. However, we cannot exclude deep minima instead of nodes. Further, a more detailed study is needed to clarify this issue.

### Details of Monte-Carlo simulations
The Monte–Carlo (MC) simulations are performed by considering a three-dimensional cubic lattice of linear dimension $L$ and lattice spacing $h = 1$. For the numerical calculations, we implement the Villain approximation[44] scheme, where the compactness of the phase

difference is ensured by writing:

$$e^{\beta \cos\left(\Delta_\mu \phi_{\mathbf{r}_i}\right)} \rightarrow \sum_{n=-\infty}^{\infty} e^{-\frac{\beta}{2}\left(\phi_{\mathbf{r}_{i+\mu}} - \phi_{\mathbf{r}_i} - 2\pi n\right)^2},$$

where $\Delta_\mu \phi_{\mathbf{r}_i} = \phi_{\mathbf{r}_{i+\mu}} - \phi_{\mathbf{r}_i}$ is the discrete phase difference between two nearest neighbors lattice sites $\mathbf{r}_{i+\mu}$ and $\mathbf{r}_i$ along $\mu = \hat{x}, \hat{y}, \hat{z}$, and $\beta = 1/T$ is the inverse temperature. The discrete Villain Hamiltonian for the model Eq. (1) in the main text reads:

$$H_v[\phi_1, \phi_2, \beta] = -\sum_{\mathbf{r}_i, \mu} \beta^{-1} \ln\left\{ \sum_{n_{1,\mu} n_{2,\mu} = -\infty}^{\infty} e^{-\beta S_\mu} \right\}, \quad (3)$$

where

$$S_\mu = \frac{1}{2}\left[u_{\mathbf{r},\mu,1}^2 + u_{\mathbf{r},\mu,2}^2\right] - \nu\left(u_{\mathbf{r},\mu,1} \cdot u_{\mathbf{r},\mu,2}\right) + \eta_2 \cos[2(\phi_{\mathbf{r},1} - \phi_{\mathbf{r},2})], \quad (4)$$

and $u_{\mathbf{r}_i,\mu,\alpha} = \Delta_\mu \phi_{\mathbf{r},j} - 2\pi n_{\mathbf{r},\mu,\alpha}$, with $\alpha = 1, 2$ label the two components. We performed MC simulations of the Villain Hamiltonian Eq. (3), locally updating the two-phase fields $\phi_1, \phi_2 \in [0, 2\pi)$ by means of the Metropolis-Hastings algorithm. A single MC step here consists of the Metropolis sweeps of the whole lattice fields while, to speed up the thermalization at lower temperatures, we implemented a parallel tempering algorithm. Typically, we propose one set of swaps after 32 MC steps. For the numerical simulations presented in this work, we performed a total of $2 \times 10^5$ Monte-Carlo steps, discarding the transient time occurring within the first 50,000 steps.

As discussed in ref. 12,14, we assessed the SC critical temperature $T_c$ by computing the helicity-modulus sum $\Upsilon_+$. That is defined as the linear response of the system with respect to a twist of the two-component phases along a given direction $\mu$:

$$\Upsilon_+^\mu = \frac{1}{L^3} \frac{\partial^2 F(\{\phi_i'\})}{\partial \hat{\delta}_\mu^2}\bigg|_{\hat{\delta}_\mu = 0}, \quad (5)$$

with: $\begin{pmatrix} \phi_1'(\mathbf{r}) \\ \phi_2'(\mathbf{r}) \end{pmatrix} = \begin{pmatrix} \phi_1(\mathbf{r}) + \delta \cdot \mathbf{r}_\mu \\ \phi_2(\mathbf{r}) + \delta \cdot \mathbf{r}_\mu \end{pmatrix}$ and $\hat{\delta}_\mu = \begin{pmatrix} \delta \cdot \mathbf{r}_\mu \\ \delta \cdot \mathbf{r}_\mu \end{pmatrix}$. In our MC simulations, the helicity-modulus sum has been computed along $\mu = \hat{x}$, so in what follows we mean $\Upsilon_+ \equiv \Upsilon_+^x$.

The critical temperature $T_c$ is then extracted by taking the thermodynamic limit of the finite-size crossings of the quantity $L\Upsilon_+$, as shown in Fig. 3a for $\nu = 0.6$ and $\eta_2 = 0.1$.

On the other hand, we extracted the critical temperature $T_c^{Z2}$ by introducing a $Z_2$ Ising order parameter $m$, equal to $+1$ or $-1$ according to the two possible signs of the ground-state phase difference $\phi_{1,2} = \pm \pi/2$. The $Z_2$ critical temperature is then determined by means of finite-size crossings of the Binder cumulant, associated with the Ising parameter $m$, extrapolated to the thermodynamic limit. The Binder cumulant $U$

$$U = \frac{\langle m^4 \rangle}{3\langle m^2 \rangle^2}, \quad (6)$$

where $\langle \ldots \rangle$ stays for the thermal average over the MC steps, is indeed expected to be a universal quantity at the critical point. The finite-size crossing points of $U$ for $\nu = 0.6$ and $\eta_2 = 0.1$ are shown in Fig. 3b.

Finally, the specific heat $C_v$ shown in Fig. 3c, d is defined as:

$$C_v = \frac{1}{T^2 L^3}\left[\langle E^2 \rangle - \langle E \rangle^2\right], \quad (7)$$

where $E$ is the total energy of the system at a given temperature $T$.

The error bars of all the observables are estimated via a bootstrap resampling method. In the figures shown, when not visible, the estimated error bars are smaller than the symbol sizes.

## Data availability

The experimental data generated in this study are provided in the Supplementary Information/Source Data file. The code generated in this study has been deposited on the Github page and is accessible with a link: https://github.com/IlaMac/MultiComponent_VillainModel. Source data are provided with this paper.

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

## Acknowledgements

E.B. was supported by the Swedish Research Council Grants 2016-06122, 2018-03659, 2022-04763. F.C. acknowledges the European Union's Horizon 2020 research and innovation program under the Marie Skłodowska-Curie grant agreement No 892728. M.P. acknowledges Italian MUR projects: PRIN 'HiBiSCUS' Grant No. 201785KWLE. I.M. acknowledges the Carl Trygger Foundation through grant number CTS 20:75. H.-H.K. acknowledges support by DFG grant SFB1143. This work was supported by a Grant-in-Aid for Scientific Research on Innovative Areas 'Quantum Liquid Crystals' (JP19H05823) from JSPS, Japan. C.-M. Y. was sponsored by Shanghai Pujiang Talent Program 21PJ1405400 and TDLI Start-up Grant. V.G. was supported by TDLI Start-up Grant. The Monte Carlo simulations were enabled by resources provided by the Swedish National Infrastructure for Computing (SNIC) at NSC partially funded by the Swedish Research Council through grant agreement no. 2018-05973. We thank Ruidan Zhong for assistance in performing experiments.

## Author contributions

I.S. performed specific heat, susceptibility, and x-ray measurements and provided interpretation of the experimental data; N.S. and F.C. performed electrical and thermal transport measurements and analyzed the data, and provided interpretation of the experimental data; I.M. wrote Monte-Carlo code, performed computer simulations and other calculations, analyzed numerical data, and co-wrote the paper; K.K. and C.H.L. prepared $Ba_{1-x}K_xFe_2As_2$ single crystals; Q.H., Y.Z. and F.Y. performed STM measurements and analyzed the data under the supervision of C.-M.Y.; Y.L. performed x-ray measurements; R.H., H.-H.K. and M.P. supervised the research and revised the manuscript; E.B. initiated and supervised the project, performed theoretical studies, provided interpretation of the experimental data and numerical results and co-wrote the paper. V.G. designed the experimental study, initiated and supervised the project, co-wrote the paper, and provided interpretation of the experimental data. All authors discussed the results and implications and commented on the manuscript.

## Competing interests

The authors declare no competing interests.
