## [Peer Review File · Nature Communications]

Calorimetric evidence for two phase transitions in $\text{Ba}_{1-x}\text{K}_x\text{Fe}_2\text{As}_2$ with fermion pairing and quadrupling statesREVIEWER COMMENTS

Reviewer #1 (Remarks to the Author):

My report is focused on the Monte Carlo simulations. The authors performed a Monte Carlo study for the Villain Hamiltonian Eq.(M1) which can be related to the model Eq.1. The simulations use the Metropolis-Hastings algorithm, which is a standard local update scheme that can be adapted to the present model. To improve the simulations at low temperatures, the parallel tempering algorithm was further employed. The lattice size is up to $L=32$, which is reasonably large, given the local update scheme of the Monte Carlo algorithm. The methodology is feasible and the results are likely to be correct. However, two issues should be addressed.

(1) The authors state that the error bars were estimated via a bootstrap resampling method. From Fig.3, it is seen that most of the error bars are visible. Relatively, the data (centers of the estimates) are connected by smooth curves. My concern is that the error bars are overestimated. Otherwise, some kinds of systematic errors were taken into account. So I invite the authors to remark on the details of estimating error bars.

(2) The finite-size scaling can be employed to analyze the universality classes of continuous phase transitions. Based on the helicity modulus (superfluid stiffness) and the Binder ratio, one can use the least-squares fits to locate the transition points and obtain the estimates of critical exponents. On this basis, the existence of U_1 and/or Z_2 transition can be checked more quantitatively.

Reviewer #2 (Remarks to the Author):

In their manuscript “Calorimetric evidence for two phase transitions in $Ba_{1-x}K_xFe_2As_2$ with fermion pairing and quadrupling states”, Ilya Shipulin et al provide new data on the $(K,Ba)Fe_2As_2$ system near 75 percent potassium substitution, where considerable evidence has accumulated that the superconducting state breaks time reversal symmetry. Specifically they report measurements of the specific heat near the superconducting transition that exhibits additional structure, not observed in previous work, that could be interpreted as two distinct anomalies, corresponding to separate phase transitions that break $U(1)$ and TRS. This observation potentially resolves one of the important lingering questions about this system, and would be a valuable contribution to our understanding of time-reversal symmetry-breaking superconductors. As such, I expect these results will be of interest to researchers in the field. However, I feel that the manuscript is not ready to be published in its current form, for reasons that I outline below.

One particularly valuable aspect of the data in this manuscript is the comparison between samples near $x=0.77$, where BTRS is observed, and samples with potassium levels outside of this region. The absence of any two-step like feature in the non-BTRS samples makes it very likely that this phenomenon is associated the Z_2 symmetry breaking. However, this leaves us with the question of why previously measured samples that show other signs of BTRS don't show two distinguishable features in the specific heat. Inhomogeneity can broaden features in the specific heat and make such a two-step structure hard to resolve. Is this what the authors believe is happening here? Do they have any results on the relative quality of these samples, either from residual resistivity values or from scanning EDS maps etc, that would substantiate that conclusion? It is notable that the three samples in which two features are discernable are at slightly lower doping (0.76, 0.75) than the original sample (S_{np} , $x = 0.77$). Do the authors think this is significant? Even among the three samples that show two anomalies, the relative size of the two anomalies varies, especially between the $x = 0.76$ and $x = 0.75$ samples, which doesn't seem natural within the authors' proposed scenario. I think a fuller discussion of the sample variation is needed to make these data convincing. One thing that could improve the presentation considerably would be some analysis of the coincidence of the superconducting signal in the magnetization with the lower-temperature feature in the specific heat. Such a coincidence seems plausible in the figures, but it depends somewhat on how the authors are defining a transition temperature in their data.

A key supporting piece of evidence in this manuscript is the observation of the spontaneous Nernst effect in sample S2. In V. Grinenko et al (Nat Phys, 2021) this effect played a major role in establishing the breaking of time reversal symmetry below the specific heat jump but above the resistive superconducting transition. In the present work however, the temperature region in which a spontaneous Nernst effect is observed does not line up nearly so well with the region of purported Z_2 symmetry breaking. The peak of the Nernst signal seems to line up with the peak of the specific heat jump and the zero of the resistive signal. This leaves half of the temperature region with a spontaneous Nernst effect sitting below the $U(1)$ symmetry breaking temperature, which seems inconsistent with scenario the authors are proposing. Furthermore, there is a kelvin or so gap between the initial rise in C/T and the first appearance of a spontaneous Nernst effect. These two features were coincident in V. Grinenko (2021), a pattern which makes more sense in the authors' proposed scenario. Here the authors seem to be associating The TRS breaking transition with a small bump around 13.25K instead of the more natural peak in $d(C/T)/dT$ just below 14K. Given the variation the authors see in the specific heat of these samples, the spontaneous Nernst data could provide essential support for the authors' conclusions, but instead the data seem to simply raise further questions.

My final area of concern is in the relationship of the Monte Carlo calculations and the authors' main conclusion. The calculations show that two distinct transitions survive in the model, and that the anomalies in the specific heat have similar magnitudes. There seems to be an implication that this model and these calculations make the experimental results reasonable. However the authors state that this phase-only model "will represent small additional contributions superimposed with the non-singular smooth contributions from pair-breaking effects". This makes it sound like model cannot for the bulk of the specific heat

signal. This is an important question because, as the authors mention in their discussion, there are important cases where there isn't a discernable specific heat feature at a BTRS transition in a superconductor (especially the case of Sr₂RuO₄). The relationship between these calculations and the experimental results should be more clearly spelled out for it to make sense to include them in this paper.

Fundamentally, the features the authors report in the specific heat are not clear or consistent enough to make a compelling case for two distinct phases on their own. Circumstantial evidence, particularly the absence of two features in samples that are far from the BTRS phase, makes it seem likely that they are related to the separation of U(1) and Z₂ symmetry breaking. Furthermore, as the authors note, the experimental relationship between BTRS and specific heat anomalies seems to be fuzzy. This means that the authors' data are potentially of significant interest to the field. However, the lack of consistency between the samples, as well as between the specific heat and Nernst data, prevents me from supporting publication of the current manuscript.

Reviewer #3 (Remarks to the Author):

Shipulin et al. has performed experiments on the Ba_{1-x}K_xFe₂As₂ material for a few values of x but focussing mainly on the x=0.75 (the so called S2 sample). They have reported the observation of two anomalies in their specific heat data for this S2 sample. The lower temperature anomaly corresponds to the transition to the superconducting state (substantiated by resistivity and susceptibility data) and the relatively higher temperature anomaly corresponds to a state with spontaneous magnetic fields (substantiated by the spontaneous data).

However, I note that all of these data and phenomenology were already reported for the x=0.77 sample in the Grinenko et al., Nat. Phys. 17, 1254 (2021), in fact possibly an anomaly (although not as clear as the x=0.75 sample reported here) in the specific heat data can already be seen there. The main result of this paper is that a little more pronounced anomaly is seen in the S2 sample specific heat data. Thus, I believe that this paper does not meet the criterion of novelty and significant advance necessary for being published in Nature Communications.

In any case, I think the authors should clarify the following points:

- 1) The authors state that "The second result of this study is the verification of the fermion quadrupling order at a different doping from the one reported in [13]": I agree that their specific heat data does indicate formation of an ordered state for the S2 case above T_c. Their interpretation that the novel state above T_c corresponds to fermion quadrupling order is consistent with the Spontaneous Nernst data.

- i) But, what experimental evidence do they have about the “fermion quadrupling” order parameter? I did not find any evidence of the formation of four fermion bound states.
- ii) Why do the authors require such an exotic type of order? Why not some type of Charge Density wave order with possible orbital loop currents developing as proposed for the case of AV_3Sb_5 (A=Cs, K, Rb) materials for example, or like Chandra Verma’s loop current state proposed for the pseudogap state in high- T_c Cuprate superconductors?
- 2) The authors use a phenomenological GL model to show that if such an exotic state is realized in this material, then the results are consistent with experiments. However, are there any material dependent calculations (microscopic model Hamiltonian based or ab-initio) to indicate that indeed such a state is expected to be realized in these materials?
- 3) What is the reason that the authors only see the pronounced 2nd anomaly in the specific heat for the $x=0.75$ sample and not so much pronounced for the other samples? The sample purity can not be the only reason as kind of indicated in the text.
- 4) What is happening to the low temperature specific heat data at $B=16T$ as reported in FIG.ED1 and Fig.2b? In particular why does it start to deviate from the phonon fit and increase with lowering temperatures?

REPLY TO THE REVIEWERS' REPORT ON THE MANUSCRIPT: "CALORIMETRIC EVIDENCE FOR TWO PHASE TRANSITIONS IN $\text{Ba}_{1-x}\text{K}_x\text{Fe}_2\text{As}_2$ WITH FERMION PAIRING AND QUADRUPLING STATES"

We would like to thank the referees for reading the manuscript and comments and very good questions that helped us to improve the paper.

REVIEWER #1

My report is focused on the Monte Carlo simulations. The authors performed a Monte Carlo study for the Villain Hamiltonian Eq.(M1) which can be related to the model Eq.1. The simulations use the Metropolis-Hastings algorithm, which is a standard local update scheme that can be adapted to the present model. To improve the simulations at low temperatures, the parallel tempering algorithm was further employed. The lattice size is up to $L=32$, which is reasonably large, given the local update scheme of the Monte Carlo algorithm. The methodology is feasible and the results are likely to be correct. However, two issues should be addressed.

We thank the referee for their careful reading of our work and for appreciating the methodology employed in the theoretical investigation of the system.

(1) The authors state that the error bars were estimated via a bootstrap resampling method. From Fig.3, it is seen that most of the error bars are visible. Relatively, the data (centers of the estimates) are connected by smooth curves. My concern is that the error bars are overestimated. Otherwise, some kinds of systematic errors were taken into account. So I invite the authors to remark on the details of estimating error bars.

We thank the referee for their critical analysis. After having submitted the paper, we have actually realized that we were overestimating the error bars of our numerical results. As the referee knows, in the bootstrap-resampling scheme the estimate of the statistical error of a given observable, say f , reads:

$$\sigma_f = \sqrt{N_B} \sigma_{f_B},$$

where $\sigma_{f_B} = \sqrt{f_B^2 - \bar{f}_B^2} / \sqrt{N_B - 1}$, N_B is the number of bootstrap resampled data sets, and \bar{f}_B is the average taken over the N_B data sets. We actually realized that, on this occasion, due to a change of coding language, we accidentally were multiplying twice σ_{f_B} by the factor $\sqrt{N_B}$, with $N_B = 100$. In the revised manuscript, we updated the figure with the correct estimate of the statistical errors.

(2) The finite-size scaling can be employed to analyze the universality classes of continuous phase transitions. Based on the helicity modulus, superfluid stiffness and the Binder ratio, one can use the least-squares fits to locate the transition points and obtain the estimates of critical exponents. On this basis, the existence of $U(1)$ and/or Z_2 transition can be checked more quantitatively.

We thank the referee for their comment and for the suggested analysis.

The estimate of the two critical temperatures relies on the finite-size scaling of the helicity modulus sum, Υ , and Binder ratio, U , for the $U(1)$ and Z_2 phase transition respectively.

As the referee highlights, from the helicity modulus sum one could in principle derive also the critical exponent ν by means of the finite-size scaling function:

$$\Upsilon L = A + BL^{1/\nu}(\beta - \beta_c) + O(L^{2/\nu}(\beta - \beta_c)^2), \quad (\text{R.1})$$

where $\beta = 1/T$.

In Fig.R1, you can see that by fitting our data with Eq. (R.1), we obtain a good fit with the critical exponent $\nu_c = 0.675 \pm 0.2$.

Despite our numerical results being in good agreement with previous estimates for the three-dimensional XY model [see for instance Ying-Hong Li and S. Teitel, Phys. Rev. B 40, 9122 (1989), A.

FIG. R1: Left panel: Finite-size scaling behaviour of the helicity modulus sum Υ_+ . The dashed grey lines are the linear fit obtained via Eq.Eq. (R.1), while the Monte Carlo numerical data correspond to different inverse temperatures, according to the colour, and different linear sizes with $L = 8, 10, 12, 20, 24, 32$. For the horizontal axis, we used $\nu_c = 0.675$. Right panel: Collapse of all the finite-size curves of Υ_+ once plotted as $L\Upsilon_+$ vs $(\beta - \beta_c)L^{1/\nu_c}$.

P. Gottlob and M. Hasenbusch, Physica A 201, 593 (1993)], a proper estimate of the critical exponent ν would require a more precise analysis, which involves different observable and larger system sizes.

In general, indeed, the assessment of the critical exponents is much more delicate than the one of the critical temperature. That is especially true when the phase transitions we are interested in characterizing are very close in temperature, like in our case. That often requires going to larger system sizes. For this reason, a proper analysis of the critical exponents for the $U(1) \times Z_2$ model goes beyond the aim of the present work, focused on the phase diagram, and we will address this study in a future publication.

REVIEWER #2

In their manuscript “Calorimetric evidence for two phase transitions in $Ba_{1-x}K_xFe_2As_2$ with fermion pairing and quadrupling states”, Ilya Shipulin et al provide new data on the (K,Ba)Fe₂As system near 75 percent potassium substitution, where considerable evidence has accumulated that the superconducting state breaks time reversal symmetry. Specifically they report measurements of the specific heat near the superconducting transition that exhibits additional structure, not observed in previous work, that could be interpreted as two distinct anomalies, corresponding to separate phase transitions that break U(1) and TRS. This observation potentially resolves one of the important lingering questions about this system, and would be a valuable contribution to our understanding of time-reversal symmetry-breaking superconductors. As such, I expect these results will be of interest to researchers in the field. However, I feel that the manuscript is not ready to be published in its current form, for reasons that I outline below.

One particularly valuable aspect of the data in this manuscript is the comparison between samples near $x=0.77$, where BTRS is observed, and samples with potassium levels outside of this region. The absence of any two-step like feature in the non-BTRS samples makes it very likely that this phenomenon is associated the Z_2 symmetry breaking. However, this leaves us with the question of why previously measured samples that show other signs of BTRS don't show two distinguishable features in the specific heat. Inhomogeneity can broaden features in the specific heat and make such a two-step structure hard to resolve. Is this what the authors believe is happening here? Do they have any results on the relative quality of these samples, either from residual resistivity values or from scanning EDS maps etc, that would substantiate that conclusion? It is notable that the three samples in which two features are discernable are at slightly lower doping (0.76, 0.75) than the original sample ($S_{np}, x = 0.77$). Do the authors think this is significant? Even among the three samples that show two anomalies, the relative size of the two anomalies varies, especially between the $x = 0.76$ and $x = 0.75$ samples, which doesn't seem natural within the authors' proposed scenario.

The question by the referee is very astute. We thank for that question. Actually for a number of subtle reasons, this behavior is precisely what we expected in our scenario: we elaborate on it below.

I think a fuller discussion of the sample variation is needed to make these data convincing. One thing that could improve the presentation considerably would be some analysis of the coincidence of the superconducting signal in the magnetization with the lower-temperature feature in the specific heat. Such a coincidence seems plausible in the figures, but it depends somewhat on how the authors are defining a transition temperature in their data.

This is also a profound question. We thank the referee for this insightful analysis. Yes, we think that even a tiny difference in the composition of the samples could significantly change the observability of the second specific heat feature. *The first reason* for that is, as well known from conventional superconductors, the phase fluctuations (i.e. beyond mean-field) contribution is very hard to detect, and it is a small feature that requires high-quality samples. The electron quadrupling state is a phase that occurs due to phase fluctuations (i.e. the phase does not exist at the mean-field level). The closest experimental example for a related specific heat problem is given by phase fluctuations in cuprates. The vortex melting transition in applied magnetic fields results in a small anomaly on the top of the large background. The size of the anomaly is very sensitive to the magnetic field and sample quality. The size of the anomaly differs significantly even for the samples with nearly the same T_c as shown in Fig. R2.

Having said that, we stress that resolving the U(1) and Z_2 transitions is even harder than resolving vortex lattice melting: the phase ordering splits into two transitions associated with interband phase

FIG. R2: Total specific heat of an untwinned $\text{YBa}_2\text{Cu}_3\text{O}_{7-\delta}$ single crystal for $H||c$. The figure is taken from ref. [Phys. Rev. Lett. **78**, 4833 (1997)] panel a and from [Science, 273(5279), 1210–1212 (1996)] panel b

differences ordering (at $T_c^{Z_2}$) and phase sum at $T_c^{U(1)}$. In our case, the samples with $x = 0.75$ (current work) and with $x = 0.77$ from [Nat. Phys. 17, 1254–1259 (2021)] have comparable quality according to electrical resistivity data. The sample $x = 0.75$ has a lower residual resistivity value $3.5 \mu\Omega\text{cm}$ compared to $4.9\mu\Omega\text{cm}$ for $x = 0.77$, but with a similar residual resistivity ratio RRR indicating that the difference in the residual resistivity is mainly related to uncertainty in the samples cross section (see Figs. R3 c and f). Nevertheless, $x = 0.77$ has a broader transition in the resistivity (panels c and f) and the susceptibility (panels a and d), which is consistent with a broader transition in the specific heat. This can be a part of the reason why the Z_2 transition is not seen for $x = 0.77$ in the zero field.

The second, even more crucial factor which affects the observability of the anomaly is the very high sensitivity of $T_c^{Z_2}$ to small variation of the doping. Hence $\Delta T_c = T_c^{Z_2} - T_c$ is also very sensitive to the small variation of the doping. Both samples phenomenologically show similar behaviour in various physical quantities but have different critical temperatures Fig. R3. The sample $x = 0.75$ has a higher T_c , defined by zero resistance, but lower $T_c^{Z_2}$, defined by the onset of spontaneous Nernst signal, as compared to the sample with $x = 0.77$. As a result, the splitting ΔT_c shrinks by nearly a factor of 3 going from $x = 0.77$ to $x = 0.75$. This effect is naturally related to a very strong doping dependence of $T_c^{Z_2}$. Obtaining $s + is$ states (or closely related $s + id$ states) requires a fine-tuning of the parameters of a generic multiband model, i.e. requires being in the parameter space between two competing “trivial” pairing symmetries. Hence, this strong dependence is consistent with a narrow dome of BTRS phase observed previously by μSR [Nat. Phys. 16, 789–794 (2020)], which is, in turn, consistent with theoretical models. It is difficult to obtain a detailed phase diagram of ΔT_c vs x due to such strong sensitivity to the sample composition. However, it is natural that $T_c^{Z_2}$ varies stronger with x near $x = 0.77$ than near $x = 0.75$. Thus, we expect that the main factor responsible for the sharpness of the anomaly at $T_c^{Z_2}$ is the rather extreme sensitivity of the temperature of the Z_2 phase to the sample composition.

The high sensitivity of $T_c^{Z_2}$ to the sample composition and quality is consistent with our theoretical description in terms of $s + is$ superconductivity. In our picture, the existence of the BTRS phase is related to “fine-tuned” frustrated interband Josephson interactions in the multiband system. This frustration can be easily lifted/diminished by small changes in doping or local strain, especially at elevated temperature.

The presence of the local doping/strain variation within the samples is evidenced by the appearance of a Meissner signal in the susceptibility at $T > T_c$. However, for both samples, the onset of the superconducting transition in the susceptibility is below $T_c^{Z_2}$ as shown in Fig. R3. This variation for $x = 0.77$ sample we studied recently using scanning SQUID as shown in Fig. R4. This allows us to conclude that the susceptibility is sensitive to local variation in the superconducting T_c . The beyond-mean-field singularities in specific heat are, in contrast, dominated by the *long-wave-lengths* fluctuations. Therefore a *local* probe would not be an appropriate quantity to define T_c in this system. Hence our specific heat signature can only be correlated with transport probes that characterize the onset of *long-range order*. From the comparison in Fig. R3 it is seen that T_c defined by zero resistance is correlated with the maximum in the specific heat for both samples.

It is important to emphasize that the size of the Z_2 anomaly, related to the contribution due to the phase difference ordering, is small for all samples compared to the non-singular contribution due to the formation of incoherent Cooper pairs (similarly to cuprates shown in Fig. R2). The seemingly varying amplitude of the Z_2 anomaly noticed by the referee is mainly related to the location of the anomaly on a broad background expected from the crossover associated with incoherent Cooper pairs formation.

A key supporting piece of evidence in this manuscript is the observation of the spontaneous Nernst effect in sample S2. In V. Grinenko et al (Nat Phys, 2021) this effect played a major role in establishing the breaking of time reversal symmetry below the specific heat jump but above the resistive superconducting transition. In the present work however, the temperature region in which a spontaneous Nernst effect is observed does not line up nearly so well with the region of purported Z_2 symmetry breaking. The peak of the Nernst signal seems to line up with the peak of the specific heat jump and the zero of the resistive signal. This leaves half of the temperature region with a spontaneous Nernst effect sitting below the $U(1)$ symmetry breaking temperature, which seems inconsistent with scenario the authors are proposing.

FIG. R3: Temperature dependencies of various physical properties measured in different magnetic fields for the single crystal with $x = 0.77$ (left) and $x = 0.75$ (right). (a, d), Temperature dependence of the zero-field specific heat $\Delta C_{e1}/T$ and the static magnetic susceptibility measured in $B||ab = 0.5$ mT. (b, e), Temperature dependence of the spontaneous Nernst effect. T_c^{Z2} is defined by the clear onset of the Nernst signal. (c, f), Temperature dependence of the electrical resistivity close to the superconducting transition. T_c is defined by zero resistance. The onset of the susceptibility for both samples falls in between T_c^{Z2} and T_c and is caused by variation of local T_c as shown below in Fig R4. The data for $x = 0.77$ sample is taken from ref. [Nat. Phys. 17, 1254–1259 (2021)].

Furthermore, there is a kelvin or so gap between the initial rise in C/T and the first appearance of a spontaneous Nernst effect. These two features were coincident in V. Grinenko (2021), a pattern which makes more sense in the authors' proposed scenario. Here the authors seem to be associating The TRS breaking transition with a small bump around 13.25K instead of the more natural peak in $d(C/T)/dT$ just below 14K. Given the variation the authors see in the specific heat of these samples, the spontaneous Nernst data could provide essential support for the authors' conclusions, but instead the data seem to simply raise further questions.

We thank the referee for this question. We would like to clarify that it is actually consistent with the scenario and also with the previous work. Since the Z_2 phase happens above $U(1) \times Z_2$ superconducting phase, the spontaneous Nernst effect is allowed by symmetry and, in general, should be present also below the superconducting transition (see Fig. R3). The Nernst signal vanishes below T_c when fluctuations are frozen, and transverse voltage is zero. Thus, it is natural to expect a maximum in the Nernst signal somewhere more or less close to T_c due to the increase of the Z_2 order parameter with the temperature decrease. The criterion that we used in both papers for T_c^{Z2} is the onset of the spontaneous Nernst signal, as shown in Figs. R3b and R3e since broken time-reversal symmetry is required for the existence of the spontaneous signal.

As the referee noticed, there are however differences at different doping level. One aspect is that the maximum for $x = 0.77$ sample is shifted to a slightly above T_c compared to $x = 0.75$. This

FIG. R4: T_c used in Figs. 2,3 are obtained as the averaged value of temperatures where the susceptibility is $-5 \Phi_0/A$ (dashed line). Inset: an optical image of the sample with scan regions. The data are taken from ref.[Science 380,1244-1247 (2023)].

can be attributed to the large difference in the splitting between $T_c^{U(1)}$ and $T_c^{Z_2}$ for two samples discussed above. The onset of broken time-reversal order happens relatively closer to superconducting phase transition for $x = 0.75$ (and hence order in the phase difference still grows significantly below superconducting critical temperature $T_c^{U(1)}$). For the sample $x = 0.77$ the $T_c^{Z_2}$ is noticeably larger than $T_c^{U(1)}$. It is essential to know the temperature of the onset of spontaneous Nernst signal, but the overall shape of the signal is, in general, not universal, and, in our case, the non-zero signal is allowed (and in general should be present) also at temperatures below $T_c^{U(1)}$ because it is allowed in the superconducting state by symmetry ($U(1) \times Z_2$) what is consistent with μ SR data.

My final area of concern is in the relationship of the Monte Carlo calculations and the authors' main conclusion. The calculations show that two distinct transitions survive in the model, and that the anomalies in the specific heat have similar magnitudes. There seems to be an implication that this model and these calculations make the experimental results reasonable. However the authors state that this phase-only model "will represent small additional contributions superimposed with the non-singular smooth contributions from pair-breaking effects". This makes it sound like model cannot for the bulk of the specific heat signal.

The referee is correct here: the reason is that there are currently no analytical or numerical methods invented to fully solve general superconducting problems beyond the mean field. In cases where mean-field is applicable, like presumably UPt_3 , one has sharp contributions. Here we look only at a tiny contribution from phase ordering, and the other contribution is relegated to a crossover/nonsingular contribution. In that respect, the situation is similar to vortex lattice melting, where one aims to resolve a small feature on top of a nonsingular background. Thus, in our specific case, we are interested in resolving the signatures in the specific heat of two phase-fluctuation-driven phase transitions, making a phase-only model we studied an appropriate model to capture essential elements of the observed physics.

This is an important question because, as the authors mention in their discussion, there are important cases where there isn't a discernable specific heat feature at a BTRS transition in a superconductor (especially the case of Sr₂RuO₄). The relationship between these calculations and the experimental results should be more clearly spelt out for it to make sense to include them in this paper.

We thank the referee for this point, we agree that it is, in general, a complicated case across all models. We rewrote the text to convey that aspect better.

Fundamentally, the features the authors report in the specific heat are not clear or consistent enough to make a compelling case for two distinct phases on their own. Circumstantial evidence, particularly the absence of two features in samples that are far from the BTRS phase, makes it seem likely that they are related to the separation of U(1) and Z₂ symmetry breaking. Furthermore, as the authors note, the experimental relationship between BTRS and specific heat anomalies seems to be fuzzy. This means that the authors' data are potentially of significant interest to the field. However, the lack of consistency between the samples, as well as between the specific heat and Nernst data, prevents me from supporting publication of the current manuscript.

We would like to thank the referee once again for raising the critical points. We realize now that we did not explain clearly in the first version that the results are consistent with theoretical expectations. Hence the referee report helped us to improve the paper. Following useful criticism, we updated the paper to address the raised issues.

REVIEWER #3

Shipulin et al. has performed experiments on the Ba(1-x)KxFe₂As₂ material for a few values of x but focussing mainly on the x=0.75 (the so called S2 sample). They have reported the observation of two anomalies in their specific heat data for this S2 sample. The lower temperature anomaly corresponds to the transition to the superconducting state (substantiated by resistivity and susceptibility data) and the relatively higher temperature anomaly corresponds to a state with spontaneous magnetic fields (substantiated by the spontaneous data).

However, I note that all of these data and phenomenology were already reported for the x=0.77 sample in the Grinenko et al., Nat. Phys. 17, 1254 (2021), in fact possibly an anomaly (although not as clear as the x=0.75 sample reported here) in the specific heat data can already be seen there. The main result of this paper is that a little more pronounced anomaly is seen in the S2 sample specific heat data. Thus, I believe that this paper does not meet the criterion of novelty and significant advance necessary for being published in Nature Communications.

We thank the referee for this critical remark that allowed us to improve the paper. What we meant by specific heat anomaly in Nat. Phys. 17, 1254 (2021) was the overall shape of the specific heat. But the puzzle and the main point of discussion during the refereeing process was the absence of any bumps at $T_c^{Z_2}$ at zero field in Nat. Phys. 17, 1254 (2021). That was the reason why we resorted to ultrasound measurements that did show a kink at $T_c^{Z_2}$ in zero external fields that only added to the puzzle. A related, although a different puzzle exists in the Sr₂RuO₄ story of trying to resolve even two-mean-field bumps (see, for example [Nat. Phys. 17, 199–204 (2021), Nat. Phys. 17, 194–198 (2021), PNAS 118 (10) e2020492118, Nature 607, 276–280 (2022)]). We would like to clarify that, we reported in Nat. Phys. 17, 1254 (2021) a field-induced kink in temperature dependencies of the specific heat close to Z_2 transition temperature defined from the spontaneous Nernst effect measurements. However the puzzle was the absence of a kink in zero fields. Hence there was a striking discrepancy between ultrasound and specific heat: this was, in fact, the main point of concern in that paper. Therefore, the observation of the specific heat anomaly in zero magnetic field at $T_c^{Z_2}$ is new and very important for characterizing BKFA.

Namely the current paper resolves that puzzle: by going to a system with smaller $T_c^{Z_2}$, we can resolve it from the nonsingular background. Moreover, the relative sizes of the bumps at $T_c^{U(1)}$ and $T_c^{Z_2}$ are consistent with our numerical simulations.

In the revised version of the manuscript, we clarify the difficulties related to resolving a separate Z_2 specific heat anomaly in a zero magnetic field.

In any case, I think the authors should clarify the following points:

1) The authors state that “The second result of this study is the verification of the fermion quadrupling order at a different doping from the one reported in [13]”: I agree that their specific heat data does indicate formation of an ordered state for the S2 case above T_c . Their interpretation that the novel state above T_c corresponds to fermion quadrupling order is consistent with the Spontaneous Nernst data.

i) But, what experimental evidence do they have about the “fermion quadrupling” order parameter? I did not find any evidence of the formation of four fermion bound states.

The evidence comes from symmetry analysis. The order in the phase difference in different bands, labeled “i,j” requires an order parameter which is constructed from four fermionic fields “c” $\langle c_i c_i^\dagger c_j^\dagger c_j \rangle$. The fact that the state is resistive means that there is no order in bilinears $\langle c_i c_i \rangle$. That is the origin of the fermion quadrupling terminology, i.e., as earlier theoretical works had shown, one cannot construct an order parameter which is second order in fermions to describe that state.

ii) Why do the authors require such an exotic type of order? Why not some type of Charge Density wave order with possible orbital loop currents developing as proposed for the case of AV3Sb5 (A=Cs, K, Rb) materials for example, or like Chadra Verma’s loop current state proposed for the pseudogap state in high- T_c Cuprate superconductors?

Initially, the main evidence came from the analysis of the superconducting state: the peculiar and detailed analysis of polarization of spontaneous magnetic fields pointed out that the low-temperature superconducting state is $s + is$ [Nat. Phys. 16, 789–794 (2020)], although similar electron quadrupling orders are equally possible in $s + id$ and other such states. I.e. the time-reversal symmetry comes from frustrated interband Josephson interaction, which in this type of material was earlier argued to occur almost inevitably at some narrow doping and was detected in some narrow doping [Nat. Phys. 16, 789–794 (2020)]. In that respect, the four-fermion order parameter $\langle c_i c_i^\dagger c_j^\dagger c_j \rangle$ already exists in the superconducting state that breaks time-reversal symmetry, along with the order in $\langle c_i c_i \rangle$. The question was whether the order $\langle c_i c_i^\dagger c_j^\dagger c_j \rangle$ disappears below or above the superconducting phase transition. The important test to establish that it is consistent with this theory was the observed suppression of both $T_c^{U(1)}$ and $T_c^{Z_2}$ by an external magnetic field, see phase diagram in Fig. 3 of ref. [Nat. Phys.17, 1254–1259 (2021)].

The referee is right that similar states may, in principle, happen in the presence of pair density wave physics (see e.g. a recent paper ref. [Phys. Rev. B 91, 054502 (2015)].) However, the analysis of spontaneous fields and the fact that our state exists in a very narrow doping range was the reason why we did not consider this possibility in Nat. Phys.17, 1254–1259 (2021).

This year other evidence was obtained that is not originating from PDW physics. In the $s + is$ scenario, the theory predicts that there should be vortices that carry a strongly temperature-dependent fraction of flux quantum. These objects were just found and reported in ref. [Science 380, 1244-1247 (2023)]. In theoretical three-band $s + is$ models $T_c^{Z_2}$ is maximal when the parent superconducting state allows fractional vortices carry 1/3 quantum of magnetic flux. The paper [Science 380, 1244-1247 (2023)] found exactly that: at $x=0.77$ fractional vortices were reaching 1/3 quantum of magnetic flux close to the transition to the fermion quadrupling state.

In any case, the referee raised an important question. To address that question directly, we performed measurements of the $x = 0.77$ sample surface using STM and found no evidence for the CDW/PDW phase. The related figure is added to the Extended Data Fig. ED3.

2) The authors use a phenomenological GL model to show that if such an exotic state is realized in this material, then the results are consistent with experiments. However, are there any material dependent calculations (microscopic model Hamiltonian based or ab-initio) to indicate that indeed such a state is expected to be realized in these materials?

Fundamentally, there are currently no theoretical methods that allow for an ab-initio demonstration of phases beyond the mean-field approximation in such fermionic models. So far full Monte-Carlo-based

inclusion of fluctuations are possible only at the level of Ginzburg-Landau or London models. However the starting model is microscopically justified. For the low-temperature state of the system various material-specific calculations support that it is multiband and $s + is$ superconductor (an overview is given in our previous works [Nat. Phys. 16, 789–794 (2020), Nat. Phys.17, 1254–1259 (2021)]). Microscopic derivation of the effects of spontaneous magnetic fields in this material, including fermion quadrupling phase, is given in Phys. Rev. Lett. 129 (8), 087602 (2022).

3) What is the reason that the authors only see the pronounced 2nd anomaly in the specific heat for the $x=0.75$ sample and not so much pronounced for the other samples? The sample purity can not be the only reason as kind of indicated in the text.

Yes, the referee is right: we agree with the referee that that is not the only reason. We discussed the different factors which can result in the smearing of the anomaly in the reply above to the second referee in detail. The important factors are the high sensitivity of the T_c^{Z2} value to the doping and the position of the anomaly relative to pair-breaking background contribution. I.e. for some samples, it is less smeared by inhomogeneities and less masked by the shape of the background.

4) What is happening to the low temperature specific heat data at $B=16T$ as reported in FIG.ED1 and Fig.2b? In particular why does it start to deviate from the phonon fit and increase with lowering temperatures?

The incoherent Cooper pair contribution to the specific heat is not completely suppressed by the 16T field. This is expected from the phase diagram shown in Fig. 3 from [Nat. Phys. 17, 1254–1259 (2021)]. We clarified this issue in the revised manuscript.

REVIEWERS' COMMENTS

Reviewer #1 (Remarks to the Author):

The authors have carefully responded to my previous report, which focused on the Monte Carlo results. The results of Monte Carlo simulations are sound, and the analyses of Monte Carlo data are now reasonable. Therefore, I am satisfied with the Monte Carlo results presented in the manuscript.

Reviewer #2 (Remarks to the Author):

I. Shipulin et al have provided a thorough reply to my critiques of their original manuscript and I now think that it is ready to be published. In particular, their discussion of T_c inhomogeneity, the associated broadness of the transitions in resistivity and specific heat, and the change of the location of the Z2 anomaly on the incoherent background makes it easier to understand why they are able to observe anomalies associated with the Z2 ordering at lower dopings. Their discussion of the theoretical relationship between the BTRS signal in the Nernst measurement and the initial rise in C/T is also improved, which is particularly helpful given their previously published data set. Overall, they have made the consistency of the full data set easier to understand, which will be helpful for the broad audience of nature communications.

I would like to make one strong suggestion about the presentation of the data in figure 1. Unless they have another reason for the order they have chosen, I suggest the authors reorder the panels in figure one according to the potassium substitution level. Noticing the evolution of the Z2 transition temperature relative to the initial rise in C/T as a function of doping is very helpful for getting a clear picture of the phenomenology that these measurements reveal.

Reviewer #3 (Remarks to the Author):

I am happy with the responses the authors have provided. I think they have carefully addressed all the points raised by me and the other referees. I therefore recommend the article for publishing.

REPLY TO THE REVIEWERS' COMMENTS ON THE MANUSCRIPT: "CALORIMETRIC EVIDENCE FOR TWO PHASE TRANSITIONS IN $\text{Ba}_{1-x}\text{K}_x\text{Fe}_2\text{As}_2$ WITH FERMION PAIRING AND QUADRUPLING STATES"

We would like to thank the referees for the positive evaluation of the manuscript.

REVIEWER #1

The authors have carefully responded to my previous report, which focused on the Monte Carlo results. The results of Monte Carlo simulations are sound, and the analyses of Monte Carlo data are now reasonable. Therefore, I am satisfied with the Monte Carlo results presented in the manuscript.

We thank the referee for carefully checking our calculations.

REVIEWER #2

I. Shipulin et al have provided a thorough reply to my critiques of their original manuscript and I now think that it is ready to be published. In particular, their discussion of T_c inhomogeneity, the associated broadness of the transitions in resistivity and specific heat, and the change of the location of the Z2 anomaly on the incoherent background makes it easier to understand why they are able to observe anomalies associated with the Z2 ordering at lower dopings. Their discussion of the theoretical relationship between the BTRS signal in the Nernst measurement and the initial rise in C/T is also improved, which is particularly helpful given their previously published data set. Overall, they have made the consistency of the full data set easier to understand, which will be helpful for the broad audience of nature communications.

I would like to make one strong suggestion about the presentation of the data in figure 1. Unless they have another reason for the order they have chosen, I suggest the authors reorder the panels in figure one according to the potassium substitution level. Noticing the evolution of the Z2 transition temperature relative to the initial rise in C/T as a function of doping is very helpful for getting a clear picture of the phenomenology that these measurements reveal.

We thank the referee for accepting our manuscript. We modified figure 1 according to the referee's recommendation.

sectionReviewer #3

I am happy with the responses the authors have provided. I think they have carefully addressed all the points raised by me and the other referees. I therefore recommend the article for publishing.

We thank the referee for accepting our revised version of the manuscript.